# COMPRESSING BERT: STUDYING THE EFFECTS OF WEIGHT PRUNING ON TRANSFER LEARNING

## ABSTRACT

Universal feature extractors, such as BERT for natural language processing and VGG for computer vision, have become effective methods for improving deep learning models without requiring more labeled data. A common paradigm is to pre-train a feature extractor on large amounts of data then fine-tune it as part of a deep learning model on some downstream task (i.e. transfer learning). While effective, feature extractors like BERT may be prohibitively large for some deployment scenarios. We explore weight pruning for BERT and ask: how does compression during pre-training affect transfer learning? We find that pruning affects transfer learning in three broad regimes. Low levels of pruning (30-40%) do not affect pre-training loss or transfer to downstream tasks at all. Medium levels of pruning increase the pre-training loss and prevent useful pre-training information from being transferred to downstream tasks. High levels of pruning additionally prevent models from fitting downstream datasets, leading to further degradation. Finally, we observe that fine-tuning BERT on a specific task does not improve its prunability. We conclude that BERT can be pruned once during pre-training rather than separately for each task without affecting performance.

## 1 INTRODUCTION

Pre-trained feature extractors, such as BERT (Devlin et al., 2018) for natural language processing and VGG (Simonyan & Zisserman, 2014) for computer vision, have become effective methods for improving the performance of deep learning models. In the last year, models similar to BERT have become state-of-the-art in many NLP tasks, including natural language inference (NLI), named entity recognition (NER), sentiment analysis, etc. These models follow a pre-training paradigm: they are trained on a large amount of unlabeled text via a task that resembles language modeling (Yang et al., 2019; Chan et al., 2019) and are then fine-tuned on a smaller amount of downstream data, which is labeled for a specific task. Pre-trained models usually achieve higher accuracy than any model trained on downstream data alone.

The pre-training paradigm, while effective, still has some problems. While some claim that language model pre-training is a "universal language learning task" (Radford et al., 2019), there is no theoretical justification for this, only empirical evidence. Second, due to the size of the pre-training dataset, BERT models tend to be slow and require impractically large amounts of GPU memory. BERT-Large can only be used with access to a Google TPU, and BERT-Base requires some optimization tricks such as gradient checkpointing or gradient accumulation to be trained effectively on consumer hardware (Sohoni et al., 2019). Training BERT-Base from scratch costs $\sim$\$7k and emits $\sim$1438 pounds of $CO_2$ (Strubell et al., 2019).

Model compression (Bucila et al., 2006), which attempts to shrink a model without losing accuracy, is a viable approach to decreasing GPU usage. It might also be used to trade accuracy for memory in some low-resource cases, such as deploying to smartphones for real-time prediction. The main questions this paper attempts to answer are: **Does compressing BERT impede it's ability to transfer to new tasks?** And **does fine-tuning make BERT more or less compressible?**

To explore these questions, we compressed English BERT using magnitude weight pruning (Han et al., 2015) and observed the results on transfer learning to the General Language Understanding Evaluation (GLUE) benchmark (Wang et al., 2019), a diverse set of natural language understanding tasks including sentiment analysis, NLI, and textual similarity evaluation. We chose magnitude

weight pruning, which compresses models by removing weights close to 0, because it is one of the most fine-grained and effective compression methods and because there are many interesting ways to view pruning, which we explore in the next section.

*Our findings are as follows*: Low levels of pruning (30-40%) do not increase pre-training loss or affect transfer to downstream tasks at all. Medium levels of pruning increase the pre-training loss and prevent useful pre-training information from being transferred to downstream tasks. This information is not equally useful to each task; tasks degrade linearly with pre-train loss, but at different rates. High levels of pruning, depending on the size of the downstream dataset, may additionally degrade performance by preventing models from fitting downstream datasets. Finally, we observe that fine-tuning BERT on a specific task does not improve its prunability or change the order of pruning by a meaningful amount.

To our knowledge, prior work had not shown whether BERT could be compressed in a task-generic way, keeping the benefits of pre-training while avoiding costly experimentation associated with compressing and re-training BERT multiple times. Nor had it shown whether BERT could be over-pruned for a memory / accuracy trade-off for deployment to low-resource devices. In this work, we conclude that *BERT can be pruned prior to distribution without affecting it's universality*, and that *BERT may be over-pruned during pre-training for a reasonable accuracy trade-off for certain tasks.*

## 2 PRUNING: COMPRESSION, REGULARIZATION, ARCHITECTURE SEARCH

Neural network pruning involves examining a trained network and removing parts deemed to be unnecessary by some heuristic saliency criterion. One might remove weights, neurons, layers, channels, attention heads, etc. depending on which heuristic is used. Below, we describe three different lenses through which we might interpret pruning.

**Compression** Pruning a neural network decreases the number of parameters required to specify the model, which decreases the disk space required to store it. This allows large models to be deployed on edge computing devices like smartphones. Pruning can also increase inference speed if whole neurons or convolutional channels are pruned, which reduces GPU usage.[1]

**Regularization** Pruning a neural network also regularizes it. We might consider pruning to be a form of permanent dropout (Molchanov et al., 2017) or a heuristic-based L0 regularizer (Louizos et al., 2018). Through this lens, pruning decreases the complexity of the network and therefore narrows the range of possible functions it can express.[2] The main difference between L0 or L1 regularization and weight pruning is that the former induce sparsity via a penalty on the loss function, which is learned during gradient descent via stochastic relaxation. It's not clear which approach is more principled or preferred. (Gale et al., 2019)

**Sparse Architecture Search** Finally, we can view neural network pruning as a type of sparse architecture search. Liu et al. (2019b) and Frankle & Carbin (2019) show that they can train carefully re-initialized pruned architectures to similar performance levels as dense networks. Under this lens, stochastic gradient descent (SGD) induces network sparsity, and pruning simply makes that sparsity explicit. These sparse architectures, along with the appropriate initializations, are sometimes referred to as lottery tickets.[3]

### 2.1 MAGNITUDE WEIGHT PRUNING

In this work, we focus on weight magnitude pruning because it is one of the most fine-grained and effective pruning methods. It also has a compelling saliency criterion (Han et al., 2015): if a weight is close to zero, then its input is effectively ignored, which means the weight can be pruned.

---

[1]If weights are pruned, however, the weight matrices become sparse. Sparse matrix multiplication is difficult to optimize on current GPU architectures (Han et al., 2016), although progress is being made.

[2]Interestingly, recent work used compression not to induce simplicity but to measure it (Arora et al., 2018).

[3]Sparse networks are difficult to train from scratch (Evci et al., 2019). However, Dettmers & Zettlemoyer (2019) and Mostafa & Wang (2019) present methods to do this by allowing SGD to search over the space of possible subnetworks. Our findings suggest that these methods might be used to train sparse BERT from scratch.

Magnitude weight pruning itself is a simple procedure: 1. Pick a target percentage of weights to be pruned, say 50%. 2. Calculate a threshold such that 50% of weight magnitudes are under that threshold. 3. Remove those weights. 4. Continue training the network to recover any lost accuracy. 5. Optionally, return to step 1 and increase the percentage of weights pruned. This procedure is conveniently implemented in a Tensorflow (Abadi et al., 2016) package[4], which we use (Zhu & Gupta, 2017).

Calculating a threshold and pruning can be done for all network parameters holistically (global pruning) or for each weight matrix individually (matrix-local pruning). Both methods will prune to the same sparsity, but in global pruning the sparsity might be unevenly distributed across weight matrices. We use matrix-local pruning because it is more popular in the community.[5] For information on other pruning techniques, we recommend Gale et al. (2019) and Liu et al. (2019b).

## 3 EXPERIMENTAL SETUP

BERT is a large Transformer encoder; for background, we refer readers to Vaswani et al. (2017) or one of these excellent tutorials (Alammar, 2018; Klein et al., 2017).

### 3.1 IMPLEMENTING BERT PRUNING

BERT-Base consists of 12 encoder layers, each of which contains 6 prunable matrices: 4 for the multi-headed self-attention and 2 for the layer's output feed-forward network.

Recall that self-attention first projects layer inputs into key, query, and value embeddings via linear projections. While there is a separate key, query, and value projection matrix for each attention head, implementations typically stack matrices from each attention head, resulting in only 3 parameter matrices: one for key projections, one for value projections, and one for query projections. We prune each of these matrices separately, calculating a threshold for each. We also prune the linear output projection, which combines outputs from each attention head into a single embedding.[6]

We prune word embeddings in the same way we prune feed-foward networks and self-attention parameters.[7] The justification is similar: if a word embedding value is close to zero, we can assume it's zero and store the rest in a sparse matrix. This is useful because token / subword embeddings tend to account for a large portion of a natural language model's memory. In BERT-Base specifically, the embeddings account for ∼21% of the model's memory.

Our experimental code for pruning BERT, based on the public BERT repository, is available here.[8]

### 3.2 PRUNING DURING PRE-TRAINING

We perform weight magnitude pruning on a pre-trained BERT-Base model.[9] We select sparsities from 0% to 90% in increments of 10% and gradually prune BERT to this sparsity over the first 10k steps of training. We continue pre-training on English Wikipedia and BookCorpus for another 90k steps to regain any lost accuracy.[10] The resulting pre-training losses are shown in Table 1.

We then fine-tune these pruned models on tasks from the General Language Understanding Evaluation (GLUE) benchmark, which is a standard set of 9 tasks that include sentiment analysis, natural

---

[4] https://www.tensorflow.org/api_docs/python/tf/contrib/model_pruning

[5] The weights in almost every matrix in BERT-Base are approximately normally distributed with mean 0 and variance between 0.03 and 0.05 (Table A). This similarity may imply that global pruning would perform similarly to matrix-local pruning.

[6] We could have calculated a single threshold for the entire self-attention layer or for each attention head separately. Similar to global pruning vs. matrix-local pruning, it's not clear which one should be preferred.

[7] Interestingly, pruning word embeddings is slightly more interpretable that pruning other matrices. See Figure 9 for a heatmap of embedding magnitudes, which shows that shorter subwords tend to be pruned more than longer subwords and that certain dimensions are almost never pruned in any subword.

[8] URL omitted for anonymity

[9] https://github.com/google-research/bert

[10] Evaluation curves leveled out at 20k steps.

language inference, etc. We avoid WNLI, which is known to be problematic.[11] We also avoid tasks with less than 5k training examples because the results tend to be noisy (RTE, MRPC, STS-B). We fine-tune a separate model on each of the remaining 5 GLUE tasks for 3 epochs and try 4 learning rates: $[2, 3, 4, 5] \times 10^{-5}$. The best evaluation accuracies are averaged and plotted in Figure 1. Individual task results are in Table 1.

BERT can be used as a static feature-extractor or as a pre-trained model which is fine-tuned end-to-end. In all experiments, we fine-tune weights in all layers of BERT on downstream tasks.

### 3.3 DISENTANGLING COMPLEXITY RESTRICTION AND INFORMATION DELETION

Pruning involves two steps: it deletes the information stored in a weight by setting it to 0 and then regularizes the model by preventing that weight from changing during further training.

To disentangle these two effects (model complexity restriction and information deletion), we repeat the experiments from Section 3.2 with an identical pre-training setup, but instead of pruning we simply set the weights to 0 and allow them to vary during downstream training. This deletes the pre-training information associated with the weight but does not prevent the model from fitting downstream datasets by keeping the weight at zero during downstream training. We also fine-tune on downstream tasks until training loss becomes comparable to models with no pruning. We trained most models for 13 epochs rather than 3. Models with 70-90% information deletion required 15 epochs to fit the training data. The results are also included in Figure 1 and Table 1.

### 3.4 PRUNING AFTER DOWNSTREAM FINE-TUNING

We might expect that BERT would be more compressible after downstream fine-tuning. Intuitively, the information needed for downstream tasks is a subset of the information learned during pre-training; some tasks require more semantic information than syntactic, and vice-versa. We should be able to discard the "extra" information and only keep what we need for, say, parsing (Li & Eisner, 2019).

For magnitude weight pruning specifically, we might expect downstream training to change the distribution of weights in the parameter matrices. This, in turn, changes the sort-order of the absolute values of those weights, which changes the order that we prune them in. This new pruning order, hypothetically, would be less degrading to our specific downstream task.

To test this, we fine-tuned pre-trained BERT-Base on downstream data for 3 epochs. We then pruned at various sparsity levels and continued training for 5 more epochs (7 for 80/90% sparsity), at which point the training losses became comparable to those of models pruned during pre-training. We repeat this for learning rates in $[2, 3, 4, 5] \times 10^{-5}$ and show the results with the best development accuracy in Figure 1 / Table 1. We also measure the difference in which weights are selected for pruning during pre-training vs. downstream fine-tuning and plot the results in Figure 3.

## 4 PRUNING REGIMES

### 4.1 30-40% OF WEIGHTS ARE NOT USEFUL

Figure 1 shows that the first 30-40% of weights pruned by magnitude weight pruning do not impact pre-training loss or inference on any downstream task. These weights can be pruned either before or after fine-tuning. This makes sense from the perspective of pruning as sparse architecture search: when we initialize BERT-Base, we initialize many possible subnetworks. SGD selects the best one for pre-training and pushes the rest of the weights to 0. We can then prune those weights without affecting the output of the network.[12]

---

[11]https://gluebenchmark.com/faq

[12]We know, however, that increasing the size of BERT to BERT-Large improves performance. This view does not fully explain why even an obviously under-parameterized model should become sparse. This may be caused by dropout, or it may be a general property of our training regime (SGD). Perhaps an extension of Tian et al. (2019) to under-parameterized models would provide some insight.

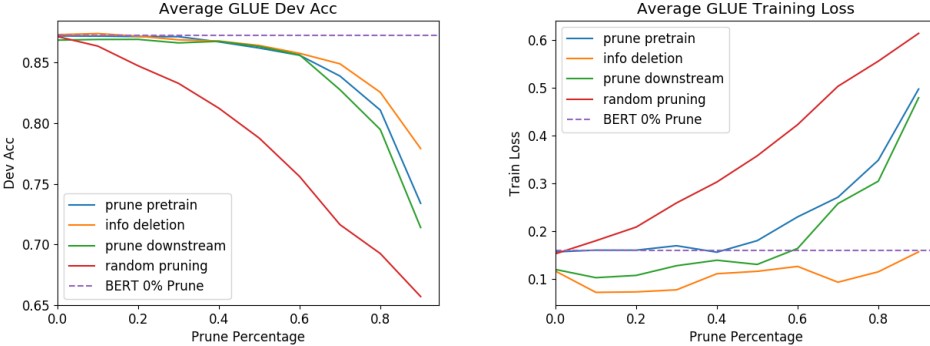

Figure 1: (Blue) The best GLUE dev accuracy and training losses for models pruned during pre-training, averaged over 5 tasks. Also shown are models with information deletion during pre-training (orange), models pruned after downstream fine-tuning (green), and models pruned randomly during pre-training instead of by lowest magnitude (red). 30-40% of weights can be pruned using magnitude weight pruning without decreasing dowsntream accuracy. Notice that information deletion fits the training data better than un-pruned models at all sparsity levels but does not fully recover evaluation accuracy. Also, models pruned after downstream fine-tuning have the same or worse development accuracy, despite achieving lower training losses. Note: none of the pruned models are overfitting because un-pruned models have the lowest training loss and the highest development accuracy. While the results for individual tasks are in Table 1, each task does not vary much from the average trend, with an exception discussed in Section 4.3.

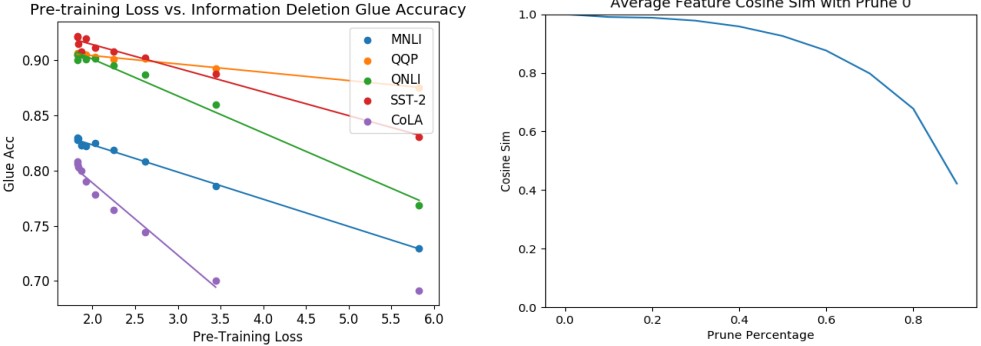

Figure 2: (Left) Pre-training loss predicts information deletion GLUE accuracy linearly as sparsity increases. We believe the slope of each line tells us how much a bit of BERT is worth to each task. (CoLA at 90% is excluded from the line of best fit.) (Right) The cosine similarities of features extracted for a subset of the pre-training development data before and after pruning. Features are extracted from activations of all 12 layers of BERT and compared layer-wise to a model that has not been pruned. As performance degrades, cosine similarities of features decreases.

## 4.2 MEDIUM PRUNING LEVELS PREVENT INFORMATION TRANSFER

Past 40% pruning, performance starts to degrade. Pre-training loss increases as we prune weights necessary for fitting the pre-training data (Table 1). Feature activations of the hidden layers start to diverge from models with low levels of pruning (Figure 2).[13] Downstream accuracy also begins to degrade at this point.

Why does pruning at these levels hurt downstream performance? On one hand, pruning deletes pre-training information by setting weights to 0, preventing the transfer of the useful inductive biases learned during pre-training. On the other hand, pruning regularizes the model by keeping certain weights at zero, which might prevent fitting downstream datasets.

Figure 1 and Table 1 show information deletion is the main cause of performance degradation between 40 - 60% sparsity, since pruning and information deletion degrade models by the same amount. Information deletion would not be a problem if pre-training and downstream datasets contained similar information. However, pre-training is effective precisely because the pre-training dataset is much larger than the labeled downstream dataset, which allows learning of more robust representations.

We see that the main obstacle to compressing pre-trained models is maintaining the inductive bias of the model learned during pre-training. Encoding this bias requires many more weights than fitting downstream datasets, and it cannot be recovered due to a fundamental information gap between pre-training and downstream datasets.[14] *The amount a model can be pruned is limited by the largest dataset the model has been trained on:* in this case, the pre-training dataset. Practitioners should be aware of this; pruning may subtly harm downstream generalization without affecting training loss.

## 4.3 HIGH PRUNING LEVELS ALSO PREVENT FITTING DOWNSTREAM DATASETS

At 70% sparsity and above, models with information deletion recover some accuracy w.r.t. pruned models, so complexity restriction is a secondary cause of performance degradation. However, these models do not recover all evaluation accuracy, despite matching un-pruned model's training loss.

Table 1 shows that on the MNLI and QQP tasks, which have the largest amount of training data, information deletion performs much better than pruning. In contrast, models do not recover as well on SST-2 and CoLA, which have less data. We believe this is because the larger datasets require larger models to fit, so complexity restriction becomes an issue earlier.

We might be concerned that poorly performing models are over-fitting, since they have lower training losses than unpruned models. But the best performing information-deleted models have the lowest training error of all, so overfitting seems unlikely.[15]

## 4.4 HOW MUCH IS A BIT OF BERT WORTH?

We've seen that over-pruning BERT deletes information useful for downstream tasks. Is this information equally useful to all tasks? We might consider the pre-training loss as a proxy for how much pre-training information we've deleted in total. Similarly, the performance of information-deletion models is a proxy for how much of that information was useful for each task. Figure 2 shows that *the pre-training loss linearly predicts the effects of information deletion on downstream accuracy.*

For every bit of information we delete from BERT, it appears only a fraction is useful for CoLA, and an even smaller fraction useful for QQP.[16] This relationship should be taken into account when

---

[13]We believe this observation may point towards a more principled stopping criterion for pruning. Currently, the only way to know how much to prune is by trial and (dev-set) error. Predictors of performance degradation while pruning might help us decide which level of sparsity is appropriate for a given trained network without trying many at once.

[14]We might consider finding a lottery ticket for BERT, which we would expect to fit the GLUE training data just as well as pre-trained BERT (Morcos et al., 2019; Yu et al., 2019). However, we predict that the lottery-ticket will not reach similar generalization levels unless the lottery ticket encodes enough information to close the information gap.

[15]We are reminded of the double-descent risk curve proposed by Belkin et al. (2018).

[16]We can't quantify this now, but perhaps compression will help quantify the "universality" of the LM task.

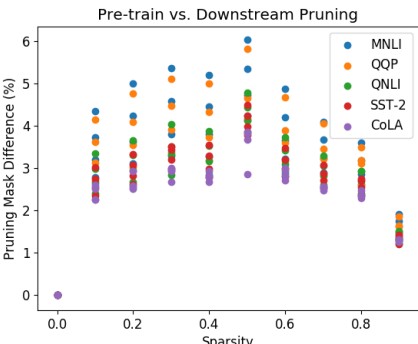 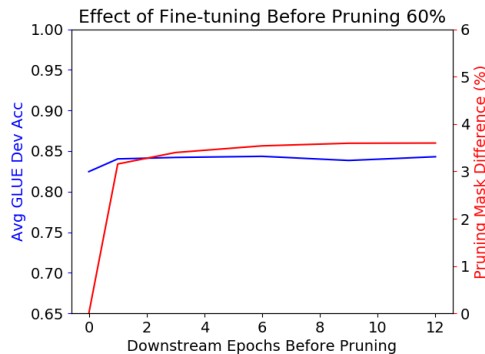

Figure 3: (Left) The measured difference in pruning masks between models pruned during pre-training and models pruned during downstream fine-tuning. As predicted, the differences are less than 6%, since fine-tuning only changes the magnitude sorting order of weights locally, not globally. (Right) The average GLUE development accuracy and pruning mask difference for models trained on downstream datasets before pruning 60% at learning rate 5e-5. After pruning, models are trained for an additional 2 epochs to regain accuracy. We see that training between 3 and 12 epochs before pruning does not change which weights are pruned or improve performance.

considering the memory / accuracy trade-off of over-pruning. Pruning an extra 30% of BERT's weights is worth only one accuracy point on QQP but 10 points on CoLA. It's unclear, however, whether this is because the pre-training task is less relevant to QQP or whether QQP simply has a bigger dataset with more information content.[17]

## 5    DOWNSTREAM FINE-TUNING DOES NOT IMPROVE PRUNABILITY

Since pre-training information deletion plays a central role in performance degradation while over-pruning, we might expect that downstream fine-tuning would improve prunability by making important weights more salient (increasing their magnitude). However, Figure 1 shows that models pruned after downstream fine-tuning do not surpass the development accuracies of models pruned during pre-training, despite achieving similar training losses. Figure 3 shows fine-tuning changes which weights are pruned by less than 6%.

Why doesn't fine-tuning change which weights are pruned much? Table 2 shows that the magnitude sorting order of weights is mostly preserved; weights move on average 0-4% away from their starting positions in the sort order. We also see that high magnitude weights are more stable than lower ones (Figure 7).

Our experiments suggest that training on downstream data before pruning is too blunt an instrument to improve prunability. Even so, we might consider simply training on the downstream tasks for much longer, which would increase the difference in weights pruned. However, Figure 4 shows that even after an epoch of downstream fine-tuning, weights quickly re-stabilize in a new sorting order, meaning longer downstream training will have only a marginal effect on which weights are pruned. Indeed, Figure 3 shows that the weights selected for 60% pruning quickly stabilize and evaluation accuracy does not improve with more training before pruning.

## 6    RELATED WORK

**Compressing BERT for Specific Tasks** Section 5 showed that downstream fine-tuning does not increase prunability. However, several alternative compression approaches have been proposed to discard non-task-specific information. Li & Eisner (2019) used an information bottleneck to discard non-syntactic information. Tang et al. (2019) used BERT as a knowledge distillation teacher to

---

[17]Hendrycks et al. (2019) suggest that pruning these weights might have a hidden cost: decreasing model robustness.

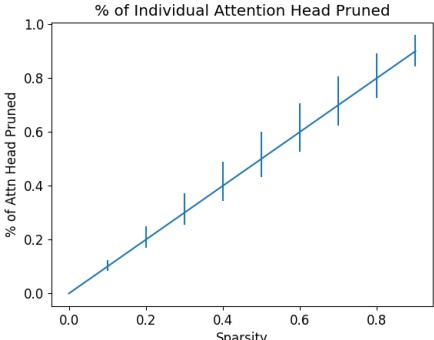 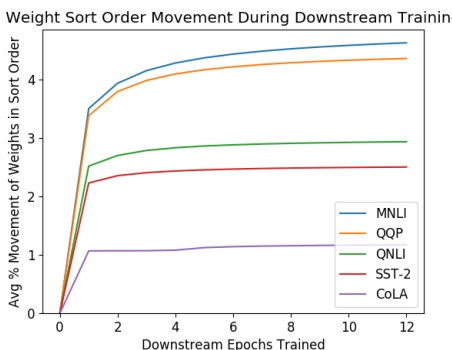

Figure 4: (Left) The average, min, and max percentage of individual attention heads pruned at each sparsity level. We see at 60% sparsity, each attention head individually is pruned strictly between 55% and 65%. (Right) We compute the magnitude sorting order of each weight before and after downstream fine-tuning. If a weight's original position is 59 / 100 before fine-tuning and 63 / 100 after fine-tuning, then that weight moved 4% in the sorting order. After even an epoch of downstream fine-tuning, weights quickly stabilize in a new sorting order which is not far from the original sorting order. Variances level out similarly.

compress relevant information into smaller Bi-LSTMs, while Kuncoro et al. (2019) took a similar distillation approach. While fine-tuning does not increase prunability, task-specific knowledge might be extracted from BERT with other methods.

**Attention Head Pruning** Voita et al. (2019) previously showed redundancy in transformer models by pruning entire attention heads. Michel et al. (2019) showed that after fine-tuning on MNLI, up to 40% of attention heads can be pruned from BERT without affecting test accuracy. They show redundancy in BERT after fine-tuning on a single downstream task; in contrast, our work emphasizes the interplay between compression and transfer learning to many tasks, pruning both before and after fine-tuning. Also, magnitude weight pruning allows us to additionally prune the feed-foward networks and sub-word embeddings in BERT (not just self-attention), which account for ∼72% of BERT's total memory usage.

We suspect that attention head pruning and weight pruning remove different redundancies from BERT. Figure 4 shows that weight pruning does not prune any specific attention head much more than the pruning rate for the whole model. It is not clear, however, whether weight pruning and recovery training makes attention heads less prunable by distributing functionality to unused heads.

## 7 CONCLUSION AND FUTURE WORK

We've shown that encoding BERT's inductive bias requires many more weights than are required to fit downstream data. Future work on compressing pre-trained models should focus on maintaining that inductive bias and quantifying its relevance to various tasks during accuracy/memory trade-offs.

For magnitude weight pruning, we've shown that 30-40% of the weights do not encode any useful inductive bias and can be discarded without affecting BERT's universality. The relevance of the rest of the weights vary from task to task, and fine-tuning on downstream tasks does not change the nature of this trade-off by changing which weights are pruned. In future work, we will investigate the factors that influence language modeling's relevance to downstream tasks and how to improve compression in a task-general way.

It's reasonable to believe that these conclusions will generalize to other pre-trained language models such as Kermit (Chan et al., 2019), XLNet (Yang et al., 2019), GPT-2 (Radford et al., 2019), RoBERTa (Liu et al., 2019a) or ELMO (Peters et al., 2018). All of these learn some variant of language modeling, and most use Transformer architectures. While it remains to be shown in future work, viewing pruning as architecture search implies these models will be prunable due to the training dynamics inherent to neural networks.

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

# A APPENDIX

| Pruned | Pre-train Loss | MNLI 392k | QQP 363k | QNLI 108k | SST-2 67k | CoLA 8.5k | AVG |
|--------|-----|-----|-----|-----|-----|-----|-----|
| 0 | 1.82 | 83.1\|0.25 | 90.5\|0.10 | 91.1\|0.12 | 92.1\|0.06 | 79.1\|0.26 | 87.2\|15.7 |
| 10 | 1.82 | 83.3\|0.21 | 90.4\|0.10 | 91.0\|0.12 | 91.6\|0.07 | 79.4\|0.30 | 87.2\|16.0 |
| 20 | 1.83 | 83.3\|0.24 | 90.5\|0.11 | 91.1\|0.11 | 91.6\|0.05 | 79.1\|0.30 | 87.1\|16.0 |
| 30 | 1.86 | 83.3\|0.23 | 90.2\|0.12 | 90.7\|0.12 | 91.9\|0.06 | 79.5\|0.31 | 87.1\|16.9 |
| 40 | 1.93 | 83.0\|0.25 | 90.1\|0.12 | 90.4\|0.12 | 91.5\|0.06 | 78.4\|0.23 | 86.7\|15.6 |
| 50 | 2.03 | 82.6\|0.27 | 89.8\|0.13 | 90.2\|0.13 | 90.9\|0.07 | 77.4\|0.30 | 86.2\|18.0 |
| 60 | 2.25 | 81.8\|0.32 | 89.4\|0.16 | 89.3\|0.16 | 91.4\|0.07 | 75.9\|0.44 | 85.6\|23.0 |
| 70 | 2.62 | 79.5\|0.40 | 88.6\|0.18 | 88.4\|0.21 | 90.1\|0.10 | 72.7\|0.47 | 83.9\|27.1 |
| 80 | 3.44 | 75.9\|0.49 | 86.9\|0.24 | 85.3\|0.29 | 88.1\|0.12 | 69.1\|0.61 | 81.1\|34.8 |
| 90 | 5.83 | 64.8\|0.76 | 81.1\|0.36 | 71.7\|0.52 | 80.3\|0.25 | 69.1\|0.61 | 73.4\|49.8 |
| | | | | Information Deletion | | | |
| 0 | 1.82 | 83.0\|0.20 | 90.6\|0.06 | 90.0\|0.10 | 92.1\|0.03 | 80.6\|0.18 | 87.3\|11.6 |
| 10 | 1.82 | 82.8\|0.01 | 90.5\|0.05 | 90.5\|0.09 | 92.2\|0.05 | 80.8\|0.16 | 87.4\|07.2 |
| 20 | 1.83 | 82.9\|0.01 | 90.5\|0.05 | 90.5\|0.09 | 91.5\|0.05 | 80.3\|0.16 | 87.2\|07.3 |
| 30 | 1.86 | 82.3\|0.01 | 90.6\|0.04 | 90.5\|0.10 | 90.8\|0.05 | 80.0\|0.18 | 86.9\|07.7 |
| 40 | 1.93 | 82.2\|0.19 | 90.5\|0.05 | 90.1\|0.10 | 92.0\|0.05 | 79.0\|0.17 | 86.7\|11.1 |
| 50 | 2.03 | 82.5\|0.19 | 90.3\|0.05 | 90.2\|0.10 | 91.2\|0.05 | 77.9\|0.19 | 86.4\|11.6 |
| 60 | 2.25 | 81.9\|0.20 | 90.1\|0.05 | 89.5\|0.10 | 90.8\|0.05 | 76.4\|0.23 | 85.7\|12.6 |
| 70 | 2.62 | 80.8\|0.01 | 90.2\|0.01 | 88.7\|0.10 | 90.3\|0.06 | 74.4\|0.28 | 84.9\|09.3 |
| 80 | 3.44 | 78.6\|0.01 | 89.3\|0.02 | 86.0\|0.02 | 88.8\|0.07 | 70.0\|0.45 | 82.5\|11.5 |
| 90 | 5.83 | 72.9\|0.01 | 87.5\|0.02 | 76.8\|0.06 | 83.0\|0.09 | 69.1\|0.61 | 77.9\|15.7 |
| | | | | Pruned after Downstream Fine-tuning | | | |
| 0 | - | 82.6\|0.15 | 90.6\|0.06 | 90.1\|0.10 | 92.1\|0.04 | 78.7\|0.25 | 86.8\|12.0 |
| 10 | - | 82.9\|0.19 | 90.6\|0.06 | 90.3\|0.10 | 91.6\|0.05 | 79.0\|0.11 | 86.9\|10.3 |
| 20 | - | 82.7\|0.15 | 90.6\|0.07 | 90.2\|0.07 | 92.0\|0.04 | 79.0\|0.22 | 86.9\|10.7 |
| 30 | - | 82.7\|0.23 | 90.4\|0.07 | 89.7\|0.07 | 91.6\|0.04 | 78.5\|0.23 | 86.6\|12.8 |
| 40 | - | 82.7\|0.25 | 90.5\|0.11 | 89.9\|0.12 | 91.7\|0.05 | 78.8\|0.17 | 86.7\|13.9 |
| 50 | - | 82.6\|0.19 | 90.3\|0.08 | 89.7\|0.11 | 90.8\|0.06 | 78.0\|0.22 | 86.3\|13.0 |
| 60 | - | 81.8\|0.22 | 90.2\|0.10 | 89.3\|0.12 | 90.6\|0.06 | 76.1\|0.31 | 85.6\|16.4 |
| 70 | - | 80.5\|0.30 | 89.4\|0.14 | 86.2\|0.19 | 88.2\|0.07 | 69.5\|0.58 | 82.7\|25.8 |
| 80 | - | 73.7\|0.53 | 87.8\|0.12 | 80.4\|0.21 | 86.4\|0.07 | 69.1\|0.59 | 79.5\|30.5 |
| 90 | - | 58.7\|0.86 | 82.5\|0.26 | 65.2\|0.52 | 81.5\|0.16 | 69.1\|0.61 | 71.4\|47.9 |
| | | | | Random Pruning | | | |
| 0 | 1.82 | 83.3\|0.26 | 90.5\|0.10 | 90.6\|0.15 | 92.4\|0.07 | 78.7\|0.18 | 87.1\|15.3 |
| 10 | 2.09 | 82.0\|0.27 | 90.1\|0.12 | 90.3\|0.13 | 92.3\|0.05 | 77.0\|0.32 | 86.3\|18.0 |
| 20 | 2.46 | 80.6\|0.32 | 89.8\|0.12 | 88.5\|0.14 | 91.1\|0.07 | 73.5\|0.39 | 84.7\|20.8 |
| 30 | 2.98 | 79.1\|0.36 | 89.2\|0.14 | 86.9\|0.23 | 89.3\|0.10 | 71.8\|0.47 | 83.3\|25.9 |
| 40 | 3.76 | 75.4\|0.45 | 88.2\|0.16 | 84.5\|0.23 | 88.6\|0.09 | 69.3\|0.57 | 81.2\|30.3 |
| 50 | 4.73 | 71.6\|0.60 | 86.6\|0.20 | 81.5\|0.28 | 85.0\|0.10 | 69.1\|0.61 | 78.8\|35.8 |
| 60 | 5.63 | 70.4\|0.60 | 85.2\|0.24 | 71.7\|0.45 | 81.5\|0.21 | 69.1\|0.61 | 75.6\|42.3 |
| 70 | 6.22 | 64.1\|0.76 | 81.4\|0.34 | 63.0\|0.62 | 80.6\|0.20 | 69.1\|0.61 | 71.6\|50.3 |
| 80 | 6.87 | 58.8\|0.84 | 76.6\|0.46 | 61.1\|0.64 | 80.6\|0.23 | 69.1\|0.61 | 69.3\|55.6 |
| 90 | 7.37 | 49.8\|0.98 | 74.3\|0.51 | 60.2\|0.65 | 75.1\|0.33 | 69.1\|0.61 | 65.7\|61.4 |

Table 1: Pre-training development losses and GLUE task development accuracies for various levels of pruning. Each development accuracy is accompanied on its right by the achieved training loss, evaluated on the entire training set. Averages are summarized in Figure 1. Pre-training losses are omitted for models pruned after downstream fine-tuning because it is not clear how to measure their performance on the pre-training task in a fair way.

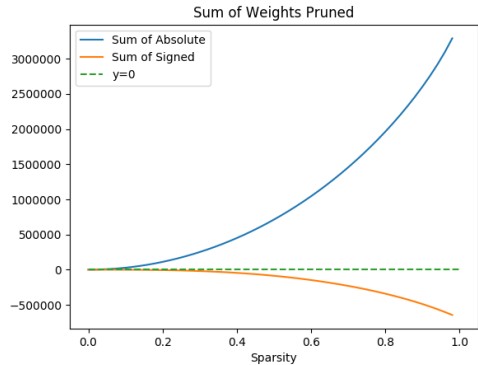

Figure 5: The sum of weights pruned at each sparsity level for one shot pruning of BERT. Given the motivation for our saliency criterion, it seems strange that such a large magnitude of weights can be pruned without decreasing accuracy.

Figure 6

| LR | MNLI | QQP | QNL | SST-2 | CoLA |
|---|---|---|---|---|---|
| 2e-5 | $1.91 \pm 1.81$ | $1.82 \pm 1.72$ | $1.27 \pm 1.22$ | $1.06 \pm 1.03$ | $0.79 \pm 0.77$ |
| 3e-5 | $2.68 \pm 2.51$ | $2.56 \pm 2.40$ | $1.79 \pm 1.69$ | $1.54 \pm 1.47$ | $1.06 \pm 1.03$ |
| 4e-5 | $3.41 \pm 3.18$ | $3.30 \pm 3.10$ | $2.31 \pm 2.19$ | $1.99 \pm 1.89$ | $1.11 \pm 1.09$ |
| 5e-5 | $4.12 \pm 3.83$ | $4.02 \pm 3.74$ | $2.77 \pm 2.62$ | $2.38 \pm 2.29$ | $1.47 \pm 1.43$ |

Table 2: We compute the magnitude sorting order of each weight before and after downstream fine-tuning. If a weight's original position is 59 / 100 before fine-tuning and 63 / 100 after fine-tuning, then that weight moved 4% in the sorting order. We then list the average movement of weights in each model, along with the standard deviation. Sorting order changes mostly locally across tasks: a weight moves, on average, 0-4% away from its starting position. As expected, larger datasets and larger learning rates have more movement (per epoch). We also see that higher magnitude weights are more stable than lower weights, see Figure 7.

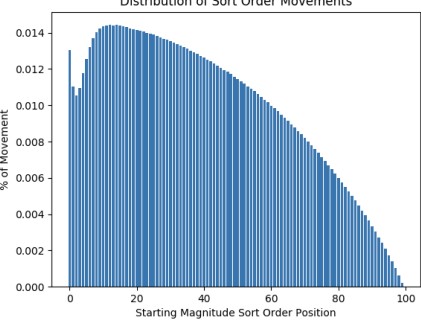

Figure 7: We show how weight sort order movements are distributed during fine-tuning, given a weight's starting magnitude. We see that higher magnitude weights are more stable than lower magnitude weights and do not move as much in the sort order. This plot is nearly identical for every model and learning rate, so we only show it once.

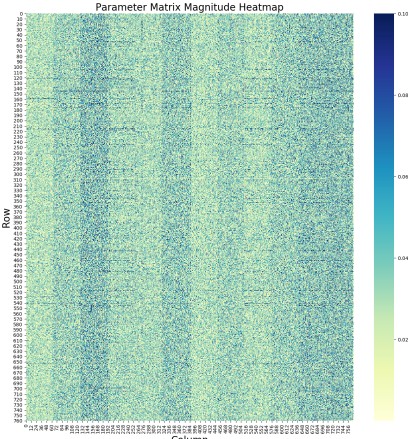

Figure 8: A heatmap of the weight magnitudes of the 12 horizontally stacked self-attention key projection matrices for layer 1. A banding pattern can be seen: the highest values of the matrix tend to cluster in certain attention heads. This pattern appears in most of the self-attention parameter matrices, but it does not cause pruning to prune one head more than another. However, it may prove to be a useful heuristic for attention head pruning, which would not require making many passes over the training data.

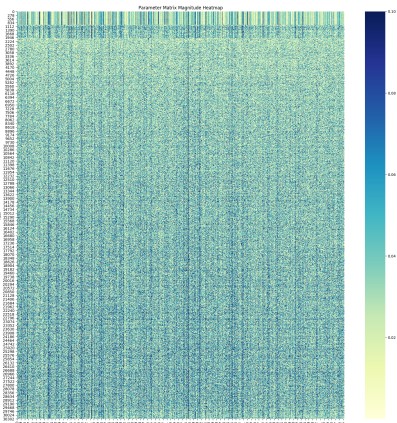

Figure 9: A heatmap of the weight magnitudes of BERT's subword embeddings. Interestingly, pruning BERT embeddings are more interpretable; we can see shorter subwords (top rows) have smaller magnitude values and thus will be pruned earlier than other subword embeddings.

| Weight Matrix | Weight Mean | Weight STD |
|---|---|---|
| embeddings word embeddings | -0.0282 | 0.042 |
| layer 0 attention output FC | -0.0000 | 0.029 |
| layer 0 self attn key | 0.0000 | 0.043 |
| layer 0 self attn query | 0.0000 | 0.043 |
| layer 0 self attn value | -0.0000 | 0.029 |
| layer 0 intermediate FC | -0.0000 | 0.037 |
| layer 0 output FC | -0.0012 | 0.036 |
| layer 1 attention output FC | 0.0001 | 0.028 |
| layer 1 self attn key | 0.0000 | 0.043 |
| layer 1 self attn query | -0.0003 | 0.043 |
| layer 1 self attn value | -0.0000 | 0.029 |
| layer 1 intermediate FC | 0.0001 | 0.039 |
| layer 1 output FC | -0.0014 | 0.038 |
| layer 10 attention output FC | -0.0000 | 0.033 |
| layer 10 self attn key | -0.0000 | 0.046 |
| layer 10 self attn query | 0.0002 | 0.046 |
| layer 10 self attn value | -0.0000 | 0.036 |
| layer 10 intermediate FC | 0.0000 | 0.039 |
| layer 10 output FC | -0.0011 | 0.038 |
| layer 11 attention output FC | -0.0000 | 0.037 |
| layer 11 self attn key | 0.0002 | 0.044 |
| layer 11 self attn query | -0.0001 | 0.045 |
| layer 11 self attn value | -0.0000 | 0.039 |
| layer 11 intermediate FC | 0.0004 | 0.039 |
| layer 11 output FC | -0.0008 | 0.036 |
| layer 2 attention output FC | 0.0000 | 0.027 |
| layer 2 self attn key | 0.0000 | 0.047 |
| layer 2 self attn query | 0.0000 | 0.048 |
| layer 2 self attn value | -0.0000 | 0.028 |
| layer 2 intermediate FC | 0.0001 | 0.040 |
| layer 2 output FC | -0.0015 | 0.038 |
| layer 3 attention output FC | 0.0001 | 0.029 |
| layer 3 self attn key | 0.0000 | 0.043 |
| layer 3 self attn query | 0.0003 | 0.043 |
| layer 3 self attn value | -0.0001 | 0.031 |
| layer 3 intermediate FC | -0.0001 | 0.040 |
| layer 3 output FC | -0.0014 | 0.039 |
| layer 4 attention output FC | 0.0000 | 0.033 |
| layer 4 self attn key | 0.0000 | 0.042 |
| layer 4 self attn query | -0.0001 | 0.042 |
| layer 4 self attn value | 0.0001 | 0.035 |
| layer 4 intermediate FC | 0.0001 | 0.041 |
| layer 4 output FC | -0.0014 | 0.040 |
| layer 5 attention output FC | -0.0000 | 0.033 |
| layer 5 self attn key | -0.0001 | 0.043 |
| layer 5 self attn query | -0.0000 | 0.043 |
| layer 5 self attn value | -0.0000 | 0.035 |
| layer 5 intermediate FC | 0.0000 | 0.041 |
| layer 5 output FC | -0.0014 | 0.039 |

| | | |
|---|---|---|
| layer 6 attention output FC | 0.0001 | 0.032 |
| layer 6 self attn key | -0.0000 | 0.043 |
| layer 6 self attn query | 0.0001 | 0.043 |
| layer 6 self attn value | 0.0000 | 0.034 |
| layer 6 intermediate FC | -0.0000 | 0.041 |
| layer 6 output FC | -0.0014 | 0.039 |
| layer 7 attention output FC | 0.0000 | 0.032 |
| layer 7 self attn key | -0.0000 | 0.044 |
| layer 7 self attn query | -0.0000 | 0.044 |
| layer 7 self attn value | 0.0001 | 0.033 |
| layer 7 intermediate FC | 0.0003 | 0.039 |
| layer 7 output FC | -0.0013 | 0.038 |
| layer 8 attention output FC | 0.0000 | 0.034 |
| layer 8 self attn key | -0.0000 | 0.044 |
| layer 8 self attn query | 0.0001 | 0.044 |
| layer 8 self attn value | 0.0000 | 0.035 |
| layer 8 intermediate FC | 0.0004 | 0.039 |
| layer 8 output FC | -0.0013 | 0.037 |
| layer 9 attention output FC | 0.0001 | 0.033 |
| layer 9 self attn key | 0.0000 | 0.046 |
| layer 9 self attn query | -0.0001 | 0.046 |
| layer 9 self attn value | 0.0000 | 0.035 |
| layer 9 intermediate FC | 0.0005 | 0.040 |
| layer 9 output FC | -0.0012 | 0.039 |
| pooler FC | 0.0000 | 0.029 |

Table 3: The values of BERT's weights are normally distributed in each weight matrix. The means and variances are listed for each.

