# OpenReview forum: "Compressing BERT: Studying the Effects of Weight Pruning on Transfer Learning"
_ICLR.cc/2020/Conference — Reject_

### Official Review · AnonReviewer1 · 2019-10-22
**Official Blind Review #1**

**Rating:** 6

**Review:**

The paper conducts a series of interesting experiments on compressing BERT and makes several conclusions. The compression technique is magnitude weight pruning based on an existing work. The paper mainly tested different compression rates and the stages when the compression can be applied. Compared to the existing work, one main contribution of the paper is to show that the BERT model can be pruned prior to fine-tuning any specific downstream tasks by 30%-40% without affecting all tested downstream tasks much. The paper is well motivated and presents interesting experimental results and conclusions. I have some concerns on their experiment details, which needs some clarification.

1. the observation in 3.4 is a little counter-intuitive to me. The model has all pre-trained weights and should be able to determine, during fine-tuning, which weights to decrease to nearly zero or to abandon. However, the experimental results show that the pruning at that point produces a worse dev accuracy. For the experiments, 3 epochs is used for fine-tuning and then the pruning is applied. I was wondering what happen if you first fine-tune the model to get the best dev accuracy and prune the weights at that point. How did you choose the number 3? I am guessing that the pruning in the middle of fine-tuning process may throw away useful information too early.
2. It will be helpful to show the thresholds of pruning and how these thresholds relate to the training loss and accuracy. I think the value of the thresholds can tell whether some pruning ratios are reasonable.
3. when the authors continue training the model, for example in 3.4, the training stops when the training losses are comparable. Why did the training loss is used as the metric instead of the dev accuracy? Figure 1 right seems to show that those models are overfitting.

**Experience Assessment:**

I have read many papers in this area.

**Review Assessment: Checking Correctness Of Derivations And Theory:**

I assessed the sensibility of the derivations and theory.

**Review Assessment: Checking Correctness Of Experiments:**

I carefully checked the experiments.

**Review Assessment: Thoroughness In Paper Reading:**

I read the paper at least twice and used my best judgement in assessing the paper.

---

> ### Author Response · Authors · 2019-11-11
> **Response to Review**
>
> Thank you for taking the time to review our paper! We will try to answer your questions in order:
>
> 1a. How did we pick 3 epochs for fine-tuning? Would a different amount of fine-tuning help?
>
> 3 epochs was the amount of fine-tuning used in the original BERT paper. However, we also tried fine-tuning between 1 and 12 epochs before pruning 60%. The results in Figure 3 (right) show that the development accuracy does not improve whether you train for 1 epoch or 12 before pruning.
>
> More importantly, however, is that the weights pruned do not change much whether we fine-tune for 1 epoch or 12 epochs. We observed in Figure 4 (right) that weights quickly settle into a new sorting order within the first epoch of fine-tuning. This implies that no matter how much longer we fine-tune, the weights selected for pruning will not change much.
>
> 1b. Why is pruning after fine-tuning worse than pruning during pre-training?
>
> This is really only true around 60% pruning and above. We believe the explanation for this is that we continue training after pruning to recover accuracy. If we prune during pre-training, this allows the model to recover some of the deleted pre-training information in the remaining weights. If we prune during fine-tuning, however, that information is no longer accessible to the model, so it cannot recover as well.
>
> 2. Would examining the pruning thresholds tell us whether some pruning ratio is reasonable?
>
> We completely agree that this is an interesting signal. It is difficult, however, to interpret the threshold without knowing the distribution of weights in each matrix. For example, a threshold of 0.01 might be reasonable if the standard deviation of weights is 1, but not if the standard deviation is 0.001.
>
> Examining the percentage of total magnitudes pruned might be a better choice here, since it is also easy to compute given the weights. This can be found in Figure 5 in the Appendix.
>
> 3a. Why were models pruned after fine-tuning trained until training losses were comparable?
>
> We wanted to make sure that the pruned models fit the downstream data just as well as models pruned during pre-training. This would imply that the models pruned after fine-tuning “learned” the downstream data just as well as the others.
>
> 3b. Doesn’t it look like the models are over-fitting?
>
> We don’t think so. The un-pruned models in each experiment have both the lowest training loss out of all the models and also the highest development accuracy. Pruning, which acts like a regularizer, decreases model complexity and increases training loss. In this case, it also happened to decrease the development accuracy.
>
> It is known that neural networks can sometimes fit the data perfectly and still generalize well. This has been called the “double-descent risk curve,”[1] and we conjecture that it may explain the role of pre-training in generalization as well as some of our results.
>
> [1] https://arxiv.org/abs/1812.11118

---

### Official Review · AnonReviewer3 · 2019-10-23
**Official Blind Review #3**

**Rating:** 3

**Review:**

This work is an empirical study of testing how pruning at the pre-training stage affects subsequent transfer learning (through fine-tuning) stage. The main idea is to carefully control the amount of sparsity injected into BERT through weight magnitude pruning and study the impact on accuracy. The experimental setup is mostly well done, especially the part that disentangles the complexity restriction and information deletion. During the exploration, the authors made several interesting observations, such as 30-40% model weights do not encode any useful inductive bias, which could help shed some light for future work on both training and compressing BERT-like models.

Overall, the paper is well written and explained. The goal is meaningful, and this is a sensible contribution to the ongoing interests of compressing BERT-like large models for efficient training and inference.

My major concern is on its novelty and how directly it can provide benefit to computation.  First, although the findings are interesting, the methods used in this paper are not new.  Various pruning techniques have been explored in prior work, which makes the novelty contribution of this paper somewhat limited.

Furthermore, the study has mostly focused on the impact of random sparsity to accuracy. However, as it is known that it is really difficult for modern hardware to benefit from random sparsity because it leads to irregular memory accesses, which negatively impact the performance. It has been observed that speedups are very limited or can be negative even the random sparsity is >95% [1]. Therefore, it is hard to judge how inference or training can benefit from 30-40% weight sparsity. Going forward, the authors are encouraged to choose pruning methods that lead to regular memory access to avoid adversely impacting practical acceleration in modern hardware platforms.

[1] Learning Structured Sparsity in Deep Neural Networks. Wen et al. NeurIPS 2016

**Experience Assessment:**

I have published one or two papers in this area.

**Review Assessment: Checking Correctness Of Derivations And Theory:**

I assessed the sensibility of the derivations and theory.

**Review Assessment: Checking Correctness Of Experiments:**

I assessed the sensibility of the experiments.

**Review Assessment: Thoroughness In Paper Reading:**

I read the paper at least twice and used my best judgement in assessing the paper.

---

> ### Author Response · Authors · 2019-11-11
> **Response to Review**
>
> Thank you so much for reviewing our paper!
>
> Our response to this review has been merged with the response to Review #2, since your concerns have a large overlap. Please view it here: https://openreview.net/forum?id=SJlPOCEKvH&noteId=HkxT-iVDor

---

### Official Review · AnonReviewer2 · 2019-10-23
**Official Blind Review #2**

**Rating:** 3

**Review:**


This work explores weight pruning for BERT. It finds that pruning affects transfer learning in three broad regimes. Low levels of pruning (30-40%) do not affect pre-training loss or transfer to downstream tasks at all. Medium levels of
pruning increase the pre-training loss and prevent useful pre-training information from being transferred to downstream tasks. High levels of pruning additionally prevent models from fitting downstream datasets, leading to further degradation.

My major concern about this work is its technical innovation and value to the community.
1. This is simply a study of model pruning for BERT. There is nothing new technically.

2. It shows BERT can be pruned for 30-40% parameters. Actually, this is not surprising; instead I'm even disappointed about this result. 30-40% weight reduction does not really speed up inference much or save model size much. Besides, to handle sparse weight matrixes, one may need additional operations to use the pruned models on a modern GPU.

3. Several other submissions show that BERT models can be compressed for 5-10x without accuracy loss. Comparing with this work, this paper seems to tell me that pruning is not suitable for BERT.

**Experience Assessment:**

I have published in this field for several years.

**Review Assessment: Checking Correctness Of Derivations And Theory:**

N/A

**Review Assessment: Checking Correctness Of Experiments:**

I assessed the sensibility of the experiments.

**Review Assessment: Thoroughness In Paper Reading:**

I read the paper at least twice and used my best judgement in assessing the paper.

---

> ### Author Response · Authors · 2019-11-11
> **Response to Review #2 and Review #3**
>
> Thank you for taking the time to review our paper! We understand your concerns
> about the practicality/novelty of the pruning method we used. To summarize:
>
> 1. Pruning does not seem practically useful.
>   a. Random sparsity is difficult to accelerate on GPUs; structured sparsity might be better.
>   b. Some methods under submission achieve higher compression rates.
>   c. We cannot prune much (30-40%) without losing accuracy.
>
> 2. We do not propose any new technical methods.
>
> We would like to emphasize, however, that these practical questions are
> orthogonal to the thesis of our paper. Our main contribution is a scientific
> investigation of a general question: how does compressing a universal feature
> extractor affect task-specific problems? We believe this paper merits being
> shared with the community on a scientific basis alone. We will, however, try to
> clarify some of the practical aspects as well.
>
> 1a. Random sparsity is difficult to accelerate on GPUs. Structured sparsity might be better.
>
> Accelerating unstructured sparse matrix multiplication is an active area of
> research in which recent progress has been made. Bank-balanced sparsity (which
> is closely related to unstructured sparsity) achieves near-ideal speed-ups while
> requiring a minimal deviation from unstructured sparsity.[1] We believe our
> results transfer to this technique, since bank-balanced sparsity preserves
> almost 95% of weight magnitudes when compared with unstructured pruning. On the
> systems side, adaptive sparse matrix multiplication has shown promising results
> on GPUs.[2]
>
> Structured pruning, on the other hand, is not currently practical. Block sparse
> pruning imposes optimization / model constraints that quickly degrade
> accuracy.[1][3] It is also not clear whether other types of structured pruning
> (attention head, etc.) are orthogonal to weight pruning (explored in Section 6),
> so it may make sense to do both on a single model.
>
> 1b. Some methods currently under submission achieve higher compression rates.
>
> These papers [4][5][6] utilize some combination of knowledge distillation, word
> embedding factorization, and parameter sharing. However, these methods are not
> well understood, which makes it difficult to use them to answer scientific
> questions. Weight magnitude pruning, on the other hand, is simple and
> well-motivated. It’s known that when an over-parameterized neural network
> achieves a global minimum, many subnetworks have zero weights [8]. We should
> also note that many of these other techniques also do not show a practical
> inference speed improvement.
>
> 1c. We cannot prune much (30-40%) without losing accuracy.
>
> As Reviewer #1 points out, 30-40% pruning is not practically useful. However,
> the specific numbers are not important to our work. Before we started, we did
> not know BERT was prunable and why performance would degrade, if it did. Our
> work has shown the existence of three distinct regimes of pruning: most of
> BERT’s capacity encodes the pre-training inductive bias, and only a small
> fraction is needed to fit downstream data. This implies that the size of the
> pre-training dataset is the limiting factor in model compression, which should
> drive future work towards understanding the nature of that inductive bias.
>
> Also, several application domains demand very memory constrained models.
> Practitioners in these domain will accept “lossy” compression (60-70%), as long
> as they can quantify the memory / accuracy trade-off.
>
> 2. We do not propose any new technical methods.
>
> We are interested in exploring the previously unexplored question of how
> compression affects transfer learning. For this purpose, we choose to use a
> well-understood technique. Weight magnitude pruning is old, but it has recently
> been validated as one of the most effective and fine-grained pruning
> techniques.[7] While other compression methods may achieve smaller model sizes,
> this is orthogonal to our main contributions.
>
> Also, some of our conclusions are independent of magnitude weight pruning:
>
> - Fine-tuning does not change the weight distribution much, giving further
>   evidence for focusing on compressing during pre-training rather than for
>   specific tasks.
>
> - Ablating BERT's inductive bias affects different tasks at different rates.
>   This provides an additional lens into why language model pre-training helps
>   other tasks, which is particularly interesting to the natural language
>   processing community, since we lack a philosophical justification for LM
>   pre-training.
>
> Again, we thank you for your reviews and hope you will consider allowing us to
> present this work at ICLR 2020.
>
> [1] https://arxiv.org/abs/1811.00206 / https://dl.acm.org/citation.cfm?doid=3289602.3293898
> [2] https://dl.acm.org/citation.cfm?doid=3293883.3295701
> [3] https://openreview.net/forum?id=HJaDJZ-0W
> [4] https://openreview.net/forum?id=H1eA7AEtvS
> [5] https://openreview.net/forum?id=rJx0Q6EFPB
> [6] https://openreview.net/forum?id=SJxjVaNKwB
> [7] https://arxiv.org/abs/1902.09574
> [8] https://tinyurl.com/yjj33x45

---

### Public Comment · ~Anonymous_Review1 · 2019-11-06
**Some Concerns about the Experiments**

1. The advantage of BERT is the BERT-LARGE, instead of BERT-BASE. This paper only shows the results on BERT-BASE, so the conclusion does not make sense to me, based only on the experiments about BERT-BASE. At least it is half-baked, maybe less than half, as the BERT-LARGE is more difficult than BERT-BASE for experiments, I think.

2. The setting about the experiments is not clear, on what kind of device do the authors conduct their experiments?

3. The pruning method in this paper is not new and is very old.

4. The final pruning ratio is just 30~40%, and the pruning process is progressive(iteratively prune 10%). I think this is drop-out, instead of pruning. Because the dropout ratio is often 20%~30%, bigger than the iteratively pruning step in this paper, and close to the final pruning result in this paper.

---

### Public Comment · ~Anonymous_Review1 · 2019-11-08
**Questions on the main claim of this paper**

Previous reviewers have mentioned that the experiment result from this paper is too limited in the perspective of weight pruning.
However, my concern is the main claim of this paper: the three regimes about transfer learning: low, medium and high. This claim maybe is right, maybe not. But the experiments in this paper do not prove their claim to me:

1, the accuracy score declines on downstream tasks do not determinedly correlates to the "prevent useful pre-training information from being transferred to downstream tasks", there are many other explanations, like this: the pruning in this paper destroyed the representation of language model. How do the authors assure the useful pre-training information is prevented to be transferred, rather than destroyed before transferred?

2, This paper discusses the transfer learning, but only adopts almost half of the tasks on GLUE, and not mentioning SQUAD. Although the authors mention their reason using one/two sentences. But this really confused me and shocked me, how the conclusion could be true on transfer leaning when you just choose half of the transfer learning tasks manually?

3, The experiment figure is coarse, in figure 1, the "average GLUE loss"? Interesting, when you choose half of the GLUE tasks, you still need to average them?

---

### Decision · Program_Chairs · 2019-12-19

**Decision:**

Reject

**Comment:**

This work explores weight pruning for BERT in three broad regimes of transfer learning: low, medium and high.

Overall, the paper is well written and explained and the goal of efficient training and inference is meaningful. Reviewers have major concerns about this work is its technical innovation and value to the community: a reuse of pruning to BERT is not new in technical perspective, the marginal improvement in pruning ratio compared to other compression method for BERT, and the introduced sparsity that hinders efficient computation for modern hardware such as GPU. The rebuttal failed to answer a majority of these important concerns.

Hence I recommend rejection.